# Seasonal Expression of Gonadotropin Genes in the Pituitary and Testes of Male Plateau Zokor (*Eospalax baileyi*)

**DOI:** 10.3390/ani12060725

**Published:** 2022-03-14

**Authors:** Kang An, Baohui Yao, Yukun Kang, Mingfang Bao, Yuchen Tan, Qiangsheng Pu, Junhu Su

**Affiliations:** 1College of Grassland Science, Gansu Agricultural University, Lanzhou 730070, China; kang_an1997@163.com (K.A.); yaobhgsau@163.com (B.Y.); yukun_kang@163.com (Y.K.); mingfang_bao@163.com (M.B.); sasuke421339218@163.com (Y.T.); p13519679121@163.com (Q.P.); 2Key Laboratory of Grassland Ecosystem, Ministry of Education, Gansu Agricultural University, Lanzhou 730070, China; 3Gansu Agricultural University—Massey University Research Centre for Grassland Biodiversity, Gansu Agricultural University, Lanzhou 730070, China

**Keywords:** plateau zokor, seasonal reproduction, *LHβ* and *FSHβ*, *LHR* and *FSHR*, RT-PCR

## Abstract

**Simple Summary:**

The plateau zokor lives on the Qinghai-Tibet Plateau and is a local endemic subterranean rodent. As a result of climate change and human activities, the population has increased rapidly, causing damage to grassland resources. Exploring reproductive mechanisms is a central issue in the effective management of the population. However, they are difficult to catch and difficult to breed artificially. Consequently, the scientific problem of seasonal reproduction of plateau zokor remains unclear. We systematically investigated the expression of gonadotropin genes in the pituitary gland and testes of the male plateau zokor during the breeding and non-breeding seasons and identified the target cells in the testes on which the luteinizing hormone and follicle stimulating hormone act. This study aims to provide a theoretical basis for further research on the seasonal reproduction of plateau zokor and explore new ideas for the effective management of their populations.

**Abstract:**

The gonadotropins, luteinizing hormone (LH) and follicle stimulating hormone (FSH), are glycoprotein hormones in the hypothalamic-pituitary-gonadal (HPG) axis and regulate mammalian reproduction. The expression of these genes in the plateau zokor (*Eospalax baileyi*) is poorly understood. We characterized the immunolocalization of the luteinizing hormone receptor (LHR) and follicle stimulating hormone receptor (FSHR) in the testes and evaluated the positive immunohistochemical results and the relative mRNA expression of gonadotropin genes. During the non-breeding season (September), the relative testes weight and the seminiferous tubule diameter were significantly reduced. All germ cell types were observed during the breeding season (May), whereas only spermatogonia were observed during the non-breeding season. LHR was present in the Leydig cells whereas FSHR was present in the Sertoli cells. The mean optical density was higher during the breeding season. The mRNA expression of *LHβ* and *FSHβ* was lower in the pituitary but *LHR* and *FSHR* genes expression were higher in the testes during the breeding season. These data elucidate the expression of gonadotropin genes in the HPG axis of the male plateau zokor and suggest that gonadotropins play a vital role in the regulation of seasonal breeding.

## 1. Introduction

Due to obvious seasonal changes in middle and high latitudes, many animals living in these areas show seasonal adaptations in behavior and physiology, such as migration and changes in reproduction and metabolism [1]. Seasonal reproduction is a self-protection strategy that emerged during the long-term evolution of animals [2]. Animals displaying seasonal reproduction can accurately predict periodic changes in environmental signals such as light, temperature, rainfall, and feed resources [3,4]. While the perception of changes in environmental signals by subterranean rodents is constrained by burrows [5], some evidence shows that there are instances of seasonal adaptation [6,7]. For instance, Herbst et al. found that an increase in testosterone concentration occurs in male Namaqua dunes mole-rats (*Bathyergus janetta*) at the beginning of the seasonal winter rainfall when the soil is moistened to facilitate the expansion of the burrow cave system, which also promotes the finding of mates [6]. In addition, the Middle East blind mole-rat (*Spalax ehrenberghi*) is still able to synchronize daily activities with the circadian cycle [7]. It is suggested that subterranean rodents may have unique and sensitive sensory systems to perceive changes in the external environment. Subsequently, they are able to make accurate judgments in their unique living environment to adjust their reproductive status.

Photoperiod is an important environmental signal that regulates mammalian seasonal reproduction [8]. Light control experiments have confirmed that gonadal gland activity in rodents is promoted by long periods of light but is inhibited by short light periods [9,10]. Light is perceived by animal eyes and the signal is transmitted into the brain to control the secretion of nocturnal melatonin by the pineal gland [11,12]. Melatonin receptor1a was found to be highly expressed in thyroid stimulating hormone (TSH), producing cells of the pars tubercles [13,14]. This indicates that melatonin may affect the synthesis and secretion of TSH. Injection of TSH or changing the photoperiod from short to long can significantly increase the expression of 2 iodothyronine deiodinases (Dio2) and the upregulation of Dio2 further increases the local thyroxine hormone (TH) concentration [15,16]. TH can induce the release of gonadotropin releasing hormone (GnRH) into the blood to control the synthesis and secretion of gonadotropic hormones in the pituitary gland [17,18].

The gonadotropins, luteinizing hormone (LH) and follicle stimulating hormone (FSH), are produced and released by the anterior pituitary cells and participate in the regulation of seasonal reproduction [19]. LH and FSH are glycoprotein hormones and comprise a common α-subunit and distinct β-subunit that confer biological specificity [20,21]. While studying LH beta subunit (*LHβ*) knockout male mice it was shown that gonadal growth and functional defects in postnatal mice lead to infertility, a decrease in testes weight, Leydig cell hypoplasia, decreased testosterone levels, and spermatogenesis in the round spermatid stage, resulting in the complete deletion of slender spermatids [22]. Zi et al. found that the expression of *LHβ* and FSH beta subunit (*FSHβ*) mRNA in the pituitary gland of prolific Lezhi black goats was higher than the levels in non-prolific Tibetan goats; this suggests that *LHβ* and *FSHβ* affect the fecundity of different goat breeds [23]. These results also suggest that the β subunit, as a component of LH and FSH, plays a key role in their physiological functions. 

LH and FSH secreted by the pituitary gland can perform biological functions only by combining with their corresponding receptors. Luteinizing hormone receptor (LHR) and follicle stimulating hormone receptor (FSHR) belong to the G-protein-coupled receptor superfamily [24]. They are the receptors of LH and FSH and exist in different animal tissues to perform different physiological functions. For example, in a study of male muskrats (*Ondatra zibethicus*), it was found that *LHR* and *FSHR* had an obvious seasonal expression pattern in the scented glands [25], suggesting that LHR and FSHR affect the development and function of muskrat scent glands.

A number of studies on LHR and FSHR have focused on animal gonadal tissue. In males, LH can promote testosterone synthesis by stimulating the Leydig cells of the testes [26]. Sertoli cells in the tubules of the testes are the target cells of FSH. The secretion of the androgen-binding protein is stimulated by the combination of FSH and FSHR in the Sertoli cells [27]. These cells have a high affinity for testosterone to maintain a high level of testosterone in the tubules of the testes which is conducive to normal spermatogenesis [27]. In addition, the combination of FSH and FSHR assists Sertoli cells in the establishment of the blood-testis barrier which is formed to separate germ cells from the external environment and provides an ideal microenvironment for spermiogenesis [28]. Therefore, the expression of gonadotropin genes in the regulation of seasonal reproduction and spermatogenesis of male animals is of great importance. 

The plateau zokor (Figure 1A) is a unique subterranean rodent resident on the Qinghai-Tibet Plateau [29,30]. It spends most of its life in underground burrows and feeds on plant roots [31]. They are native species of the alpine ecosystem, regarded as ecosystem engineers, and they play an active role in maintaining the alpine ecosystem [32]. Recently, the secretion pattern of serum LH and FSH hormone levels in the plateau zokor during the breeding and non-breeding seasons has been revealed [33]. Moreover, the reproductive activity of the plateau zokor is obviously cyclical, with adult females giving birth once a year between April and July [34]. This evidence indicates that the plateau zokor is a typical seasonally reproductive species. Therefore, the seasonally secreted gonadotropins play a key role in the regulation of reproduction in the plateau zokor. We hypothesized that the expression of gonadotropin genes is responsible for the seasonal functions of LH and FSH. However, systematic studies of gonadotropin genes expression in the plateau zokor are limited. This study explored the immune localization of gonadotropin receptors in testicular tissue and the expression of gonadotropin genes during the breeding and non-breeding seasons of the plateau zokor. We aimed to reveal the expression of gonadotropin genes in the male plateau zokor which is of great significance for understanding the regulation in seasonal reproduction of this subterranean rodent.

## 2. Materials and Methods

### 2.1. Animals and Tissues

We used an automatic live catching cage (Baoji Ludixincheng Co., Ltd., Xi’an, China) to capture male plateau zokor, in May (*n* = 10, average monthly temperature = 5.5 °C; day length = 13.14 h) and September (*n* = 10, average monthly temperature = 6.4 °C; day length = 12.66 h) 2020, located in Zhuaxixiulong Town, Tianzhu County in Gansu Province, China. Adult males with a body weight of 250–350 g were used in the experiments. The selection was based on the maturity of the plateau zokor [35]. Individuals with normal growth and development were selected, and the pituitary and testes were extracted rapidly after euthanizing the plateau zokors; both testes were collected and weighed. One testis was fixed using a 4% paraformaldehyde tissue fixative for histological and immunohistochemical testing. The pituitary gland and the other testis were promptly placed in liquid nitrogen and then stored at −80 °C for later analysis to determine the relative expression levels of gonadotropin genes. The Animal Ethics Committee of Gansu Agricultural University approved the experimental procedure and the local authorities approved our research (GAU-LC-2020-014).

### 2.2. HE Staining

Testicles of appropriate size were obtained from the 4% paraformaldehyde stored samples. Samples were examined from the breeding (*n* = 10) and non-breeding (*n* = 10) seasons. First, they were placed in an embedding box and rinsed with water for 24 h to remove any remaining fixative. After the embedding was complete, 5 µm sections were sliced using a rotary slicer (Leica RM2255, Beijing, China). The slices were respectively stained with hematoxylin for 15 s, extracted 2–3 times with 0.1% hydrochloric acid, and dyed in 1% eosin solution for 5 s. Finally, the slices were dehydrated and sealed with neutral gum before microscopic examination and photography. The resulting slides were observed and compared under a microscope (Panthera U, Motic^®^, Xiamen, China). The number of Sertoli cells was counted during the breeding season and non-breeding seasons [36]. 

### 2.3. Immunohistochemistry

The paraffin sections of testicular tissue were prepared in an oven at 60 °C for 6 h in order to melt excess paraffin at high temperature and prepare for subsequent full dewaxing. After conventional dewaxing and washing with distilled water, these slices were soaked in sodium citrate solution (pH 6.0) and boiled twice, with an intermediate pause of 3 min for antigen repair, and 3% H_2_O_2_ was used to eliminate endogenous peroxidase activity. Next, these slices were sealed with normal goat serum working fluid to reduce the non-specific absorption of the detected antibodies. The experimental group was treated with rabbit anti-LHR (Beijing Boaosen, BS-6431R, 1:250) and rabbit anti-FSHR (Beijing Boaosen, BS-2065R, 1:250), and the negative control was treated with PBS. All samples were stored overnight at 4 ℃. Sheep anti-rabbit IgG and horseradish enzyme-labeled chain albumen working solution was then added, incubated at 37 °C, and stained with DAB (Beijing Boaosen, C-0010). One slice from each experimental animal (breeding season: *n* = 10; non-breeding season: *n* = 10) was selected. Motic Images Plus 3.0 software was used to randomly select three different visual fields for photography. Image Pro Plus (MediaCybernetics^®^, Silver Spring, MD, USA) was used to detect the immunopositive mean optical density of LHR and FSHR in testicular tissue during both the breeding and non-breeding seasons. 

### 2.4. Total RNA Isolation and First-Strand cDNA Synthesis

Total RNA was obtained using TRIzol (RNAiso Plus^®^, Takara, Beijing, China) according to the manufacturer’s instructions. Utilizing nuclease-free water to elute RNA, genomic DNA was removed by DNase treatment (recombinant DNase I^®^, Takara, Beijing, China). After DNase treatment, the DHS NanoPro 2010^®^ spectrophotometer (DHS Technologies, Inc., Beijing, China) was used to quantify the RNA. The A_260_/A_280_ values of all samples were higher than 1.8, indicating that the RNA quality was high. Samples were stored at −80 °C for subsequent experiments. Furthermore, cDNA synthesis was performed by reverse transcription of 1 μg of total RNA using PrimeScript 1st Strand^®^ (Takara) following the manufacturer’s instructions. The obtained DNA (cDNA) was stored at −20 °C for subsequent work. 

### 2.5. Primer Design

The reference sequences were obtained from the published sequence of *LHβ* (XM_029569375.1), *FSHβ* (XM_008851216.1), *LHR* (XM_021253842.1) and *FSHR* (XM_021253847.1) in naked mole-rat (*Heterocephalus glaber*) on NCBI (https://www.ncbi.nlm.nih.gov/nuccore/, accessed on 20 June 2021). The primers were designed using Primer3Input (Version 0.4.0) software after sequence comparison with MEGA7. A commercial sequencing system (TsingKe, Xi’an, China) was used to synthesize the primer sequences (Table 1). The Xi’an TsingKe Biotechnology Company designed the *β-actin* sequence. 

### 2.6. RT-PCR

The cDNA was amplified by real-time PCR (qPCR) using TB Green^®^ Premix Ex Taq™ II (Takara, 9767/9767S) in a real-time PCR system (Light Cycler 96 System, Roche Life Science, Penzberg, Germany). The total reaction volume was 20 µL and there were three replicates for each sample. The reaction consisted of TB Green Premix Ex Taq II, 10 µL; cDNA template (500 µg/mL), 2 µL; forward primer (10 um/µL), 1 µL; reverse primer (10 um/µL), 1 µL; and RNase free water, 6 µL. The reaction conditions were as follows: one cycle at 95 °C for 30 s, 40 cycles at 95 °C for 5 s, and 60 °C for 30 s; one cycle of 5 s at 95 °C and 60 s at 60 °C and one cycle for 30 s at 50 °C. Melting curve analysis showed that the amplified PCR products of *LHβ*, *FSHβ*, *LHR*, *FSHR,* and *ACTB* were single. We constructed a standard curve for each gene using a five-fold series of cDNA dilutions. To guarantee the reliability of the relative quantitative method, we analyzed the standard curves of target genes and *ACTB* and the results showed that they have similar amplification efficiencies.

### 2.7. Statistical Analysis

Motic Images Plus software (version 3.0) was used to take photos of the results of HE staining and immunohistochemistry experiments. Image J was used to measure the seminiferous tubule diameter in testicular tissue during different stages of HE staining. Image-Pro Plus was used to evaluate the average optical density. The 2^−ΔΔCq^ method was used to deal with the results of RT-PCR, and the relative expression level of each gene was corrected using the reference gene [37]. All data were analyzed using SPSS 26.00 software using an independent sample *t*-test. The mean ± standard error (mean ± SE) was used to represent the place of the results and the data were statistically plotted using Excel.

## 3. Results

### 3.1. Histology and Morphology

There were significant morphological differences in testicular tissue between breeding and non-breeding seasons (Figure 1B). The relative testes weight and the seminiferous tubule diameter were lower during the non-breeding season (*p* < 0.05) (Figure 1C,D). During the breeding season, many different germ cells were observed in the seminiferous tubules, including spermatogonia, primary spermatocytes, secondary spermatocytes, and elongate spermatids (Figure 1E). However, during the non-breeding season, only Sertoli cells and spermatogonia were observed (Figure 1F). The proportion of Sertoli cells during the breeding season and the non-breeding season was 42.67% and 27.83%, respectively. 

### 3.2. Immunohistochemical Results

#### 3.2.1. Localization of LHR and FSHR

In the Leydig cells of testes tissue during the breeding season, we observed a strong expression of LHR but only a weak signal in the Sertoli cells and germ cells in the seminiferous tubules. The immune signal of LHR was stronger in the Leydig cells than that in the seminiferous tubule cells of the testes during the non-breeding season (Figure 2A,D). During the breeding season, FSHR was strongly expressed in the Sertoli cells of the tubules of testicular tissue. However, it was only weakly expressed in the Leydig cells and the cytoplasm of germ cells. It was not observed in elongate spermatids. In non-breeding testicular tissue, FSHR was positively expressed only in the Sertoli cells and not in the Leydig cells and spermatogonia (Figure 2B,E).

#### 3.2.2. Immunopositive Evaluation of LHR and FSHR

It was found that the average optical density of the LHR and FSHR immunopositive signal was similar and there was a difference between seasons (*p* < 0.01). During the breeding season, the average optical density of LHR and FSHR immunopositive signals was significantly higher (Figure 3A,B).

### 3.3. RT-PCR Analysis

#### 3.3.1. The Gene Expression in the Pituitary

The mRNA expression of *LHβ* and *FSHβ* in the pituitary gland was detected during both the breeding and non-breeding seasons. During the non-breeding season, the expression of *LHβ* and *FSHβ* was higher (*p* < 0.05; Figure 4A). The PCR products of the primers were sequenced and compared with the nucleotide sequence homology of rats, mice, marmots, and humans (Table 2). The plateau zokor *LHβ* cDNA nucleotide sequence homology with rats, mice, marmots, and humans was 83.7%, 82.7%, 84.5%, and 73.0%, respectively, and 84.1%, 85.7%, 87.4%, and 83.6%, for *FSHβ,* respectively (Table 2).

#### 3.3.2. The Gene Expression in Testes

The mRNA expression of *LHR* and *FSHR* was detected in the testicular tissue of the plateau zokor. The results corresponded with the results of the average optical density values. There were significant differences between *LHR* and *FSHR* expression between the breeding and non-breeding seasons; the relative expression level was higher during the breeding season (*p* < 0.05; Figure 4B). The plateau zokor *LHR* cDNA nucleotide sequence homology with the sequences of rats, mice, marmots, and humans was 92.7%, 95.2%, 90.1%, and 88.0%, respectively; and 88.2%, 85.9%, 90.4%, and 87.7%, respectively for *FSHR* (Table 2).

## 4. Discussion

Although the plateau zokor is a typical subterranean rodent, its reproductive activity still shows an obvious seasonal pattern [34]. During the non-breeding season, we found that the relative testes weight and the seminiferous tubule diameter decreased significantly. Several studies have indicated that the testicular tissue of seasonally reproductive animals undergoes periodic changes throughout the year, including changes in testicular size, spermatogenesis, testosterone synthesis, and secretion [9,38,39]. This phenomenon has been confirmed in studies of specific species such as the golden hamster (*Mesocricetus auratus*), giant panda (*Ailuropoda melanoleuca*), and deer (*Capreolus capreolus*) [9,39,40]. Injections of LH can stimulate the proliferation and differentiation of testicular Leydig cells [41]; additionally, the number and mean volume of Leydig cells per testicle tripled in adult rats treated with LH for two weeks [42]. Furthermore, the addition of FSH can increase the number of cultured Sertoli cells and round sperm cells in vitro [43]. Similarly, injecting FSH into rats can prevent gonad degeneration accompanied by an increase in the Sertoli cells and an increase in testicular volume [44]. These studies explain that LH and FSH play key roles in the growth and development of testicular tissue. Consequently, we can infer that the periodic changes in testicular tissue of plateau zokor are due to the seasonal secretion of gonadotropin by the pituitary, which causes periodic cell proliferation by binding to receptors on target cells. Moreover, we also found that LHR was mainly located in the Leydig cell of the testicular tissue of the plateau zokor and FSHR was mainly present in the Sertoli cells. In the 1980s, Wahlström et al. studied the location of LHR and FSHR in the testes of humans and rats by immunohistochemistry [45]. They found that LHR was positively expressed in the Leydig cells and FSHR was positively expressed in the Sertoli cells of both species [45]. Similar results were also obtained in wild ground squirrels (*Citellus dauricus*), where LHR immunopositivity was localized to the Leydig cells of testicular tissue, while FSHR was detected in the Sertoli cells [46]. These results indicate that the adjustment of LHR and FSHR in testicular tissue is mediated by different types of cells. Recently, An et al. found that LH and FSH are secreted seasonally during plateau zokor spermatogenesis and levels are high during the breeding season [33]. We can infer that LH and FSH secreted by the pituitary at high levels combine with their corresponding receptors to promote the testicular development in the plateau zokor during the breeding season. In contrast, low levels of secreted hormones restrict cell proliferation and tissue development during the non-breeding season. As a result, the testicular tissue undergoes periodic changes throughout the year.

Interestingly, we found that the expression of *LHβ* and *FSHβ* genes were higher in the pituitary of plateau zokor during the non-breeding season, which was contrary to the changes in the relative testes weight and the seminiferous tubule diameter. Similar results were found in the Libyan jird (*Meriones libycus*), where *LHβ* and *FSHβ* mRNA expression was higher during the autumn non-breeding season, which is contrary to changes in the testes, seminal vesicle weight, and plasma testosterone [47]. When Siberian hamsters (*Phodopus sungorus*) were transferred from an inhibitory short photoperiod to a stimulating long photoperiod, the mRNA levels of *FSHβ* increased significantly after 5 d, then decreased, but remained higher than the short photoperiod levels [21]. The results suggest that when the expression level of the *FSHβ* gene increases to a certain level, there may be a feedback mechanism to inhibit its continued high-level expression. Further studies have found that serum FSH decreased in long-term castrated golden hamsters after inhibin treatment while serum LH did not change [48]. This indicates that inhibin may hinder the expression of the *FSHβ* gene through a feedback mechanism, producing a decrease in serum FSH. Moreover, summer gonadectomy increased *LHβ* and *FSHβ* mRNA levels in newts (*Cynops pyrrhogaster*), while testosterone replacement inhibited *LHβ* mRNA expression but not *FSHβ* mRNA expression [49]. This indicates that testosterone is a repressor of the *LHβ* gene repression. In this study, we did not detect the hormone levels of inhibin and testosterone in the serum of the plateau zokor, but we observed that the expression of *LHβ* and *FSHβ* genes in the pituitary were contrary to the seasonal changes in testicular tissue. This suggests that inhibin and testosterone may have affected the expression of *LHβ* and *FSHβ* genes. We will investigate this in subsequent experiments.

In the current study, we observed that the relative mRNA expression levels of *LHR* and *FSHR* in the testes of plateau zokor were higher during the breeding season. In a study of wild ground squirrels’ testicular tissue, it was found that *LHR* and *FSHR* genes were expressed at a higher level during the breeding season [46]. The activation of a series of adenylate cyclase/protein kinase sequences when LH binds to LHR regulates steroid production in the Leydig cells, promotes the synthesis of testosterone and estradiol, and facilitates animal reproduction [50]. In *LHR* knockout mice, it was found that although the number of Sertoli cells and spermatogonia in the testes was not different from that in the control group, round sperm cells were significantly reduced and elongated spermatids were completely absent [51], indicating that the *LHR* gene has a direct effect on spermatogenesis. 

Sertoli cells provide nutrition and protection for germ cells [52], which determines the number of germ cells supported by spermatogenesis [53]. Furthermore, meiosis proceeds normally when mammalian spermatogenic cells are co-cultured with Sertoli cells in vitro [54,55], indicating that Sertoli cells play an important role in spermatogenesis. It was found that testosterone alone did not restore spermatogenic populations if chronic FSH treatment was discontinued in gonadotropin-immune rats [56]. In addition, male mice without FSHR, produced by homologous recombination technology, demonstrated fertility although the testes were smaller and some spermatogenesis failed [44], indicating that FSH and FSHR also play a direct role in spermatogenesis.

The tight connections between Sertoli cells form the blood-testis barrier which separates germ cells from the external environment and provides a stable internal environment for spermatogenesis [28]. In vitro studies have shown that when FSH is increased, adjacent adhesive junctions of Sertoli cells are fused to form extensive adhesive bonding bands, showing that FSH and FSHR can stimulate Sertoli cells to form bonding bands facilitating the establishment of a blood-testosterone barrier and demonstrating that FSHR may have an indirect role in spermatogenesis [57]. Therefore, we inferred that *LHR* and *FSHR* genes are highly expressed in the testes of plateau zokor during the breeding season and high levels of receptor proteins are translated in the Leydig cells and Sertoli cells, which, combined with the gonadotropins secreted by the pituitary gland, may promote spermatogenesis and facilitate reproduction in the plateau zokor. However, low levels of genes expression during the non-breeding season restricts testicular development and spermatogenesis so that the plateau zokor can store more energy to withstand adverse living conditions.

## 5. Conclusions

The present study found that the testicular size of plateau zokor has obvious periodic changes that are similar to those of aboveground seasonal reproductive animals. Moreover, we identified the target cell types of LH and FSH in the testes of the plateau zokor and systematically elucidated the expression of gonadotropin genes in the HPG axis of the male plateau zokor during the different breeding seasons. We can infer that gonadotropin plays an indispensable role in the periodic changes in testes and the regulation of seasonal reproduction. These results provide a scientific reference for understanding the seasonal reproduction of animals, especially subterranean rodent species.

## Figures and Tables

**Figure 1 animals-12-00725-f001:**
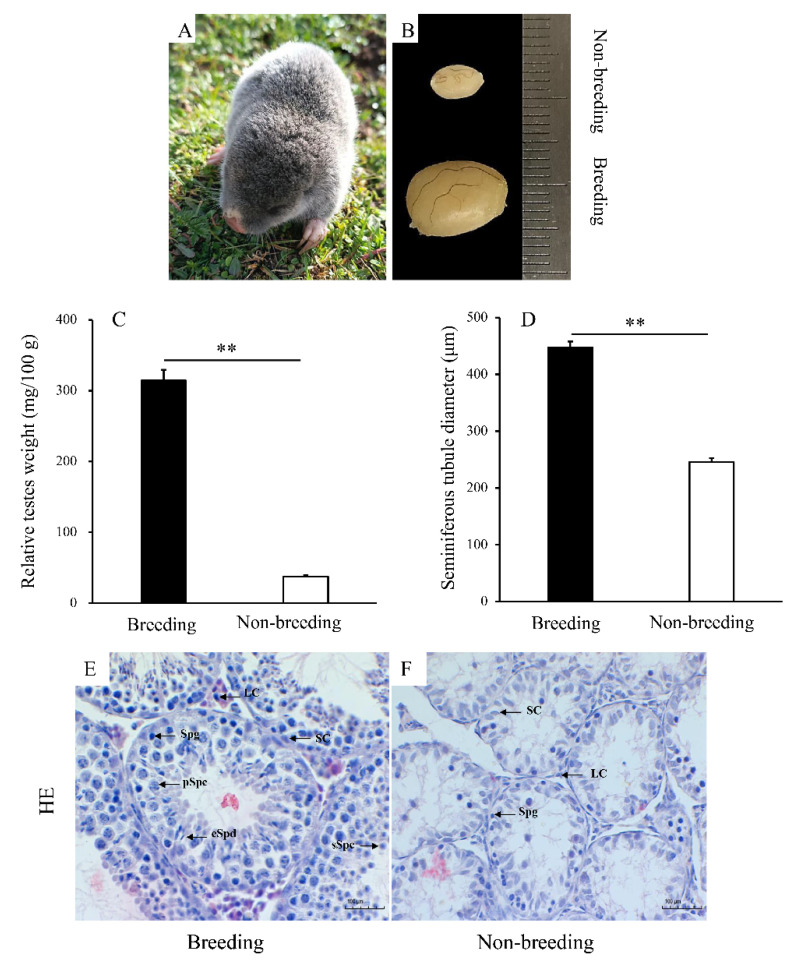
The testicular morphology, weights, and histology of plateau zokor (*Eospalax baileyi*) during the transition from the breeding to the non-breeding season. (**A**): the plateau zokor, (**B**): the testes after fixation, (**C**): the relative testes weight of plateau zokor during the breeding and the non-breeding seasons, (**D**): the seminiferous tubule diameter of plateau zokor during the breeding and the non-breeding seasons, and (**E**,**F**): the histological change of the testes from the breeding season (**E**) to the non-breeding season (**F**). Scale bars, 100 µm. SC, Sertoli cell; LC, Leydig cell; Spg, spermatogonia; pSpc, primary spermatocyte; sSpc, secondary spermatocyte; eSpd, elongated spermatid. Bars represent the means ± SD of two independent experiments (*n* = 10 per season). Means within the columns with different marks indicate a significant difference; ** represents *p* < 0.01.

**Figure 2 animals-12-00725-f002:**
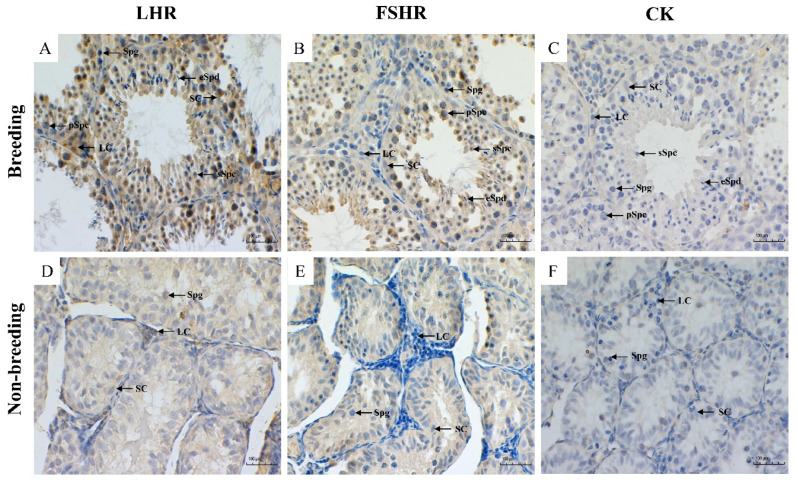
Immunohistochemistry results of luteinizing hormone receptor (LHR) and follicle stimulating hormone receptor (FSHR) in the testes of plateau zokor (*Eospalax baileyi*) during the breeding and non-breeding seasons. (**A**,**D**): the immune localization of LHR in the testes of plateau zokor, (**B**,**E**): the immune localization of FSHR in the testes of plateau zokor, and (**C**,**F**): the control using phosphate buffered saline (PBS) instead of the first antibody. Scale bars, 100 µm. SC, Sertoli cell; LC, Leydig cell; Spg, spermatogonia; pSpc, primary spermatocyte; sSpc, secondary spermatocyte; eSpd, elongated spermatid.

**Figure 3 animals-12-00725-f003:**
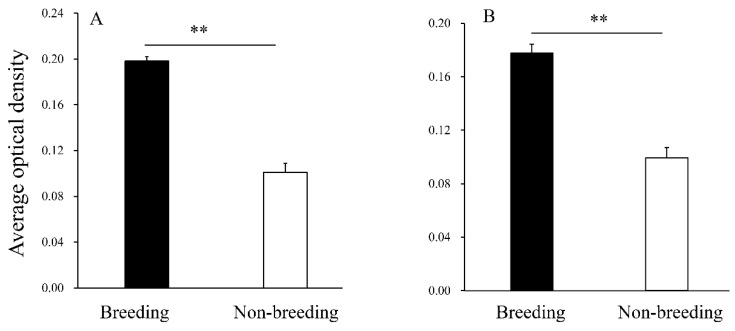
Average optical densities of immunopositive signals of luteinizing hormone receptor (LHR) and follicle stimulating hormone receptor (FSHR) during the breeding and non-breeding seasons. (**A**): average optical densities of immunopositive signals of LHR; (**B**): average optical densities of immunopositive signals of FSHR. Bars represent means ± standard deviations of two independent experimental treatments (*n* = 10 per season). Means within columns with different marks indicate a significant difference (** represents *p* < 0.01).

**Figure 4 animals-12-00725-f004:**
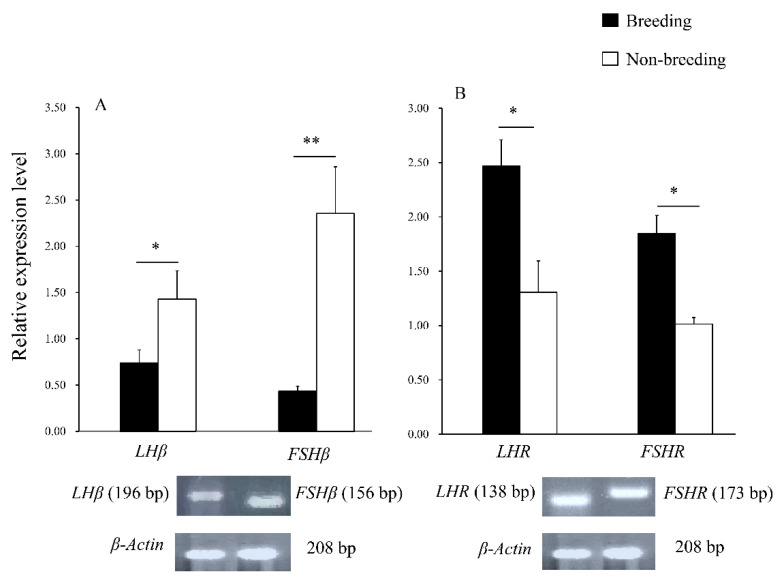
RT-PCR analysis of the mRNA of plateau zokor (*Eospalax baileyi*) during the breeding and non-breeding seasons; (**A**): the mRNA relative expression levels of *LHβ* and *FSHβ* in the pituitary gland of plateau zokor (**B**): the mRNA relative expression levels of *LHR* and *FSHR* in the testes of plateau zokor. Bars represent the means ± standard deviation of two independent experimental treatments (*n* = 10 per season). Means within the columns with different marks indicate a significant difference; * represents *p* < 0.05, ** represents *p* < 0.01.

**Table 1 animals-12-00725-t001:** Primer sequences of *LHβ*, *FSHβ*, *LHR,* and *FSHR* of plateau zokor (*Eospalax baileyi*) for the investigation of the expression of gonadotropin in the pituitary and testes.

Genes	Primer Sequence	Temperature (°C)	Size (bp)
*LHβ*	F: 5′-CAACCCTGGCTGCAGAGAAT-3′	53.5	196
	R: 5′-GGGCCACAGGAAAGGAGAC-3′		
*FSHβ*	F: 5′-CAGGCTACTGCTACACCAGG-3′	53.5	156
	R: 5′-CAGTGGCTACTGGGTACGTG-3′		
*LHR*	F: 5′-ATGCCTTTGACAACCTCCTCA-3′	54	138
	R: 5′-GGGTCTGGATGCCTGTGTTA-3′		
*FSHR*	F: 5′-GCTGAGGCCTTCCAGAATCTT-3′	54	173
	R: 5′-AAACTCAGTCCCATGAAGGAAT-3′		
*β*–Actin	F: 5′-TTGTGCGTGACATCAAAGAG-3′	53.5	208
	R: 5′-ATGCCAGAAGATTCCATACC-3′		

**Table 2 animals-12-00725-t002:** Comparison of nucleotide sequence homology between the plateau zokor (*Eospalax baileyi*) and rats (*Rattus rattus*), mice (*Mus musculus*), marmots (*Marmota marmota*), and humans (*Homo sapiens*).

Gene Name	Rat (%)	Mouse (%)	Marmot (%)	Human (%)
*LHβ*	83.7	82.7	84.5	73.0
*FSHβ*	84.1	85.7	87.4	83.6
*LHR*	92.7	95.2	90.1	88.0
*FSHR*	88.2	85.9	90.4	87.7

## Data Availability

Data and associated calculation tools are available from the author upon reasonable requestion kang_an1997@163.com.

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
