# Peer review of "Seasonal Expression of Gonadotropin Genes in the Pituitary and Testes of Male Plateau Zokor (Eospalax baileyi)"

_animals, 2022, doi:10.3390/ani12060725_

Round 1

Reviewer 1 Report

Seasonal expression of gonadotropin genes in the pituitary and testes of male plateau zokor (Eospalax baileyi)

In my opinion the authors investigated an interesting topic with could have practical impact on control of an expanding species. The introduction prepares the reader well and sufficient for understanding the following. In my opinion the hypothesis behind the investigation should be expressed more clearely. Concerning the materials an method part I was misseng a detailed description how the optical density of the immunohistology was measured (It was metioned in part 3.2.2. but should be moved to 2! In the RT-PCR part (2.6) the absolute concentartions of the reagents (primer etc.) should be indicated (Molarities!). 117: ...were used...

In my opinion the number of Sertoli cells during the breeding or non-breeding season could be crucial for alterations in spermiogenesis may be the authors still have access to evaluate this numbers in order to improve the value of the manuscript (see also: 

Front. Endocrinol., 14 December 2018 | https://doi.org/10.3389/fendo.2018.00763

Role of Follicle-Stimulating Hormone in Spermatogenesis

Olayiwola O. Oduwole1, Hellevi Peltoketo2 and Ilpo T. Huhtaniemi1,3*

Author Response

 We are very grateful to your comments for the manuscript. Our responses to the comments are provided below.

1 In my opinion the hypothesis behind the investigation should be expressed more clearly.

Response: Thank you for your comment. The relevant instructions have been added to the revised article (Lines 106-109).

2 Concerning the materials an method part I was missing a detailed description how the optical density of the immunohistology was measured (It was mentioned in part 3.2.2. but should be moved to 2!

Response: Thank you for your pertinent suggestion. We have made the necessary changes (Lines 158-160).

3 In the RT-PCR part (2.6) the absolute concentrations of the reagents (primer etc.) should be indicated (Molarities!). 117: ...were used...

Response: Thank you for your suggestion. We have made the necessary changes (Lines 186-187).

4 In my opinion the number of Sertoli cells during the breeding or non-breeding season could be crucial for alterations in spermiogenesis may be the authors still have access to evaluate this numbers in order to improve the value of the manuscript (see also:

Front. Endocrinol., 14 December 2018 | https://doi.org/10.3389/fendo.2018.00763

Role of Follicle-Stimulating Hormone in Spermatogenesis

Olayiwola O. Oduwole1, Hellevi Peltoketo2 and Ilpo T. Huhtaniemi1,3*

Response: Thank you for your pertinent suggestion. According to Yin et al. (1998), we counted the number of Sertoli cells during breeding and non-breeding seasons (Table R1). The relevant results have been added to the revised article (Lines 140-142; 212-213).

Table R1. The number of Sertoli cells during the breeding and non-breeding seasons of plateau zokor in this study.

Period

Sample number

Total number of cells

Total number of Sertoli cells

Sertoli cells ratio

Breeding

10

10000

4267

42.67%

Non-breeding

10

10000

2783

27.83%

Yin, Y.; Demolf, W.C.; Morgentaler, A. Experimental cryptorchidism induces testicular germ cell apoptosis by p53-dependent and independent pathways in mice. Biol. Reprod. 1998, 58, 492–496.

Reviewer 2 Report

The MS “Seasonal expression of gonadotropin genes in the pituitary and 2 testes of male plateau zokor (Eospalax baileyi)” describes differences in testis size and morphology in males of the plateu zokor betwenn May (breeding) and September no breeding). Results are of interest and I think probably the work was well done, however there is an uncorrected use of English that difficult the reading and understanding of the MS.

- Authors use the term “expression pattern” to describe differences only between May and September, I think only two point is not a pattern (a seasonal pattern). Please modify this through all the text.

- Material and methods section is not clear, maybe because language is not correctly used, or even because some important details are missing:

Please include data of temperature and photoperiod in the days of May and September that animals were captured.

Lines 128-129, 10 testes per group? (total 20 testes?)

Lines 120-121. Which is the size of one testis? It is small enough to be well fixed just by immersion in PAF?

Line 131, “using a machine” please specify which type of cryostat was used.

Line 132. “Finally, hematoxylin and eosin were used to stain the sections” Please give more details or a reference.

Line 136, please give more details. “was successively sliced and prepared in an oven at 60 °C for 6 h”.

Line 141, I think that the “control group” is the control of the immunohistochemistry or a negative control, but an experimental group and a control group?

Line 140, I think that the word “antibody” should be included before LHR and FSHR. How were these antibodies tested? For each species were they synthesized? They were specific for rat? Mouse?

Lines 137-139, “After routine dewaxing and rinsing with distilled water, the antigen was repaired with sodium citrate solution, incubated with 3% H2O2 to eliminate endogenous peroxidase activity, and the serum was blocked.”

I do not understand antigen was repaired?

The serum was used to block I imagine…

Lines 143-144. “Six slides were selected for each period and each slide was photographed with three different visual fields under a microscope. ”How these six slides were selected? Six of each animal? Authors indicate that they used 10 animals in each season, were these 10 animals analysed for all studied parameters? Please indicate this clearly.

Line 155, please include the quantity of RNA used and not the volume.

Minor comments:

Lines 79-77. What authors want to say with “These studies indicate that LHβ and FSHβ play an indispensable role in the seasonal breeding of mammals”? Obviously, with no sexual hormones there is no reproduction.

Line 77. Begin the new paragraph with “LH and FSH secreted by the pituitary gland only combine with their 77 corresponding receptors to perform biological functions”

Line 99, something is missing “These animals an endemic”.

There are several language errors. I noted here some examples but there are more:

line 116-117. “Adult males with a 116 body weight of 250–350 g was used in the experiments.”  “Was” should be substituted by “were”, please revise English through all the text.

Lines 169-170. Something is missing in this sentence. “To amplify the cDNA by real-time PCR (RT-PCR) in a real-time PCR system (Light 169 Cycler 96 System, Roche Life Science) with TB Green® Premix Ex Taq™ II (Takara).”

Author Response

We are very grateful to your comments for the manuscript. According with your advice, we amended the relevant part in manuscript. Some of your questions were answered below.

1 Authors use the term “expression pattern” to describe differences only between May and September, I think only two point is not a pattern (a seasonal pattern). Please modify this through all the text.

Response: Thank you for your valuable comment. We have changed the “expression pattern” to “expression”. We have highlighted the changes made using red text in our revised manuscript.

- Material and methods section is not clear, maybe because language is not correctly used, or even because some important details are missing:

2 Please include data of temperature and photoperiod in the days of May and September that animals were captured.

Response: Thank you for pointing this out. The relevant instructions have been added to the revised article (Lines 118-120).

3 Lines 128-129, 10 testes per group? (total 20 testes?)

Response: Thank you for your comment. There were 10 testes during the breeding season and 10 testes during non-breeding season, and the total number of testes in this study was 20. The relevant instructions have been added to the revised article (Line 133).

4 Lines 120-121. Which is the size of one testis? It is small enough to be well fixed just by immersion in PAF?

Response: Thank you for your comment. Figure 1B showed the size of testes, and the weight of testes during the breeding and non-breeding season was 0.98 ± 0.04 g and 0.12 ± 0.01 g, respectively (from the original data). Therefore, PAF treatment can be used for subsequent experiment.

5 Line 131, “using a machine” please specify which type of cryostat was used.

Response: Thank you for pointing this out. We have made the necessary changes (Line 136).

6 Line 132. “Finally, hematoxylin and eosin were used to stain the sections” Please give more details or a reference.

Response: Thank you for pointing this out. The relevant instructions have been added to the revised article (Lines 136-139).

7 Line 136, please give more details. “was successively sliced and prepared in an oven at 60 °C for 6 h”.

Response: Thank you for pointing this out. The relevant instructions have been added to the revised article (Lines 144-146).

8 Line 141, I think that the “control group” is the control of the immunohistochemistry or a negative control, but an experimental group and a control group?

Response: We apologize for the wrong description in our manuscript. The experimental group was treated with rabbit anti-LHR (Beijing BoaoSen, BS–6431R, 1: 250) and rabbit anti-FSHR (Beijing BoaoSen, BS–2065R, 1: 250) as primary antibodies respectively, while the negative control group was treated with PBS instead of primary antibodies. All the groups were stored overnight at 4℃. The relevant instructions have been added to the revised article (Lines 152-153).

9 Line 140, I think that the word “antibody” should be included before LHR and FSHR. How were these antibodies tested? For each species were they synthesized? They were specific for rat? Mouse?

Response: Thank you for your comment. These antibodies were synthesized by Beijing BoaoSen Technology Co., Ltd., and expressed as rabbit anti-LHR (Beijing Boao Sen, BS–6431R, 1: 250) and rabbit anti-FSHR (Beijing Boao Sen, BS–2065R, 1: 250). They were specific for rat (Rattus norvegicus). The relevant instructions have been added to the revised article (Lines 151-152).

10 Lines 137-139, “After routine dewaxing and rinsing with distilled water, the antigen was repaired with sodium citrate solution, incubated with 3% H2O2 to eliminate endogenous peroxidase activity, and the serum was blocked.”

10.1 I do not understand antigen was repaired?

Response: Thank you for your comment. The relevant instructions have been added to the revised article (Lines 146-148).

10.2 The serum was used to block I imagine…

Response: Thank you for your comment. The relevant instructions have been added to the revised article (Lines 149-150).

11 Lines 143-144. “Six slides were selected for each period and each slide was photographed with three different visual fields under a microscope.” How these six slides were selected? Six of each animal? Authors indicate that they used 10 animals in each season, were these 10 animals analyzed for all studied parameters? Please indicate this clearly.

Response: We apologize for the wrong description in our manuscript. Six slides at least in each season were used. In the actual experiment process, one slice of each experimental animals was used (Breeding season: n = 10; non-breeding season: n = 10). We have made the necessary changes. (Lines 155-157).

12 Line 155, please include the quantity of RNA used and not the volume.

Response: Thank you for pointing this out. The relevant instructions have been added to the revised article (Line 169).

13 Lines 79-77. What authors want to say with “These studies indicate that LHβ and FSHβ play an indispensable role in the seasonal breeding of mammals”? Obviously, with no sexual hormones there is no reproduction.

Response: Thank you for pointing this out. We have made the necessary changes (Lines 76-78).

14 Line 77. Begin the new paragraph with “LH and FSH secreted by the pituitary gland only combine with their 77 corresponding receptors to perform biological functions”

Response: Thank you for pointing this out. We have made the necessary changes (Lines 79-80).

15 Line 99, something is missing “These animals an endemic”.

Response: Thank you for pointing this out. The descriptions here have been changed (Lines 100-101).

There are several language errors. I noted here some examples but there are more:

16 line 116-117. “Adult males with a 116 body weight of 250–350 g was used in the experiments.” “Was” should be substituted by “were”, please revise English through all the text.

Response: Thank you for pointing this out. We have made the necessary changes (Line 121).

17 Lines 169-170. Something is missing in this sentence. “To amplify the cDNA by real-time PCR (RT-PCR) in a real-time PCR system (Light 169 Cycler 96 System, Roche Life Science) with TB Green® Premix Ex Taq™ II (Takara).”

Response: Thank you for pointing this out. We have made the necessary changes (Lines 183-185).
